# A previous hemorrhagic stroke protects against a subsequent stroke via microglia alternative polarization

Shin-Shin Lee[1], Li Pang[1], Yin Cheng[1], Jia Xin Liu[1], Anson Cho Kiu Ng[1] & Gilberto Ka Kit Leung [1✉]

Microglia in hemorrhagic stroke contribute to both acute-phase exacerbation and late-phase attenuation of injury. Here, by using the mouse model, we reported that the shift in polarization of microglia from M1 to M2 phenotype could be altered by a past 'mini' stroke, resulting in better neurological function recovery, faster attenuation of lesion volume, and better survival. In mice with a previous stroke, M2 predominance appeared markedly in advance compared to mice without a previous stroke. Mechanistically, the RBC-mediated M2 polarization of microglia was synergistically enhanced by T cells: microglia cocultured with RBCs alone resulted in mild alterations to M2 markers, whereas in the presence of T cells, they expressed an early and sustained M2 response. These results suggest that by harnessing the microglia-mediated M2 polarization response, we could help mitigate devastating sequelae before a prospective hemorrhagic stroke even happens.

[1] Department of Surgery, LKS Faculty of Medicine, The University of Hong Kong, Hong Kong, SAR, China. ✉email: gilberto@hku.hk

Hemorrhagic stroke (HS) refers to an intracerebral rupture of blood vessels leading to injury to the brain[1]. Spontaneous HS due to hypertension most commonly occurs in the putamen and caudate nucleus (50–60%)[1]. The overall prognosis is poor, with a 30-day mortality rate of 35–52%[2]. One-half of these deaths occur within the first two days[2], stressing the importance of acute brain injury in pathogenesis. The mechanisms of injury within the first few hours include the hematoma mass effect and ensuing toxicity of red blood cell (RBC) components. The release of free hemoglobin, heme, and iron, followed by the hemolytic cascade, triggers a second wave of damage after the initial bleed[3]. As blood enters the normally blood-free brain parenchyma, reactive immune responses commence. The dormant immune cells within the brain, microglia, respond to injury within minutes to release chemokines and cytokines, propagating the inflammatory process and secondary brain insults[4].

Microglia migrate caudally during embryonic development to stay within the brain parenchyma throughout life[5]. They are similar to bone marrow-derived macrophages morphologically and functionally, with the ability to phagocytose and present foreign antigens to T cells via major histocompatibility complex (MHC) class II molecules (MHC-II). Microglia, comprising 10–20% of the total cells within the brain, also surveil the cerebral immune environment and contribute to many major neuroinflammatory actions of various diseases such as ischemic stroke, multiple sclerosis, traumatic brain injury, and infections[6,7].

Following an HS, microglia first polarize via the classic polarization pathway, resulting in a phenotypic spectrum commonly deemed as M1 microglia[8]. This group is characterized by its proinflammatory actions that contribute to perilesional edema and exacerbate a cerebral injury. As time progresses, another polarization pathway, which results in a phenotype broadly classified as M2 microglia starts to assume dominance[9]. M2 microglia comprises a wide spectrum of cells that promotes cell regeneration (M2a), phagocytosis of hemolytic products (M2b and M2c), and removal of cell debris (M2b and M2c). This shift in polarization from M1 to M2 normally takes place on day 7[9]. Although the dichotomization of microglia into either M1/M2 phenotype is overgeneralized, for the sake of simplicity we would refer to them as such within this study.

In the present study, we explored an incidental finding: mice that had undergone two separate episodes of HS performed significantly better on functional studies when compared with those without a previous HS. This phenomenon has never been discussed previously, and we aimed to determine the mechanisms underlying the above observation. Interestingly, we also observed the modulatory effects of a previous HS on subsequent strokes for the first time. Behavioral studies showed a significant trend, so we further performed MRI imaging, demonstrating attenuation of total lesion volume in animals with a previous HS. A recent study has put forth a concept of priming stimuli as mice that received multiple peripheral lipopolysaccharide injections showed attenuation of inflammation when exposed to Alzheimer's disease and ischemic stroke[10]. As microglia are responsible for removing hematoma and cell debris[11,12], we subsequently performed molecular experiments relating to microglia function and found an altered polarization pattern of microglia within the subacute phase. Thus, our study provided phenotypical and mechanistic insight into the phenomenon that may facilitate future research and clinical treatment of successive hemorrhagic strokes.

## Results

**Mice with a previous stroke had a lower mortality rate and exhibited faster and better functional recovery after a subsequent stroke**. Mice with a previous stroke demonstrated superior survival rates compared to the control group (Fig. 1b). There was a significant decrease in body weight initially in both groups on day 1 post-HS. From day 3 on, the control group exhibited a further decrease in body weight, while the previous stroke group stabilized. On day 5, both control and previous stroke groups exhibited recovery of lost body weight, but the average weight of the previous stroke group was still higher than that of the control. The difference in body weight leveled out on day 7 (Fig. 1c). During mNSS assessments, both the control and treatment groups showed a high severity score on day 1 post-stroke. However, on day 3, the control group demonstrated significantly more severe neurologic deficits compared to that of the previous stroke group. This difference continued through days 5 and 7, with the previous stroke group recovering at a higher rate and to a greater extent compared to that of the control group (Fig. 1d). Rotarod testing revealed that, on day 1, both groups had substantial difficulties remaining on the rod and dropped within 50 s of test commencement. On day 3, the difference between the two groups became apparent. The control group was unable to stay on the accelerating rod for as long as the previous stroke group, indicating an inferior recovery of motor function. This disparity persisted to days 5 and 7, with the difference in latency-to-fall increasing between the two groups (Fig. 1e).

**Lesion resolution occurred at a higher rate in mice with a previous stroke**. Total hematoma volume was quantified by histological sectioning. The two groups had a similar initial hematoma size, as demonstrated in Fig. 2a, b on day 1, progressing on to a more rapid clearance in hematoma volume on days 3 and 5 in the previous stroke group, with the difference between the two groups decreasing on day 7 (Fig. 2c). MRI imaging, although unable to provide absolute delineation of hematoma borders, provided us with a total lesion volume consisting of a hematomal core, peri-hematomal edema, and surrounding rim. MRI scans revealed equivalent total lesion volumes on day 1 post-stroke in both groups (Fig. 3a), indicating equal amounts of bleeding and a stable surgical model. Major differences in total lesion volumes became apparent on days 3 and 5 (Fig. 3b), with the decline of the previous stroke group approximately tripling that of the control: mean total lesion volume being 63 and 93% on day 3 in the previous stroke group and control group respectively, and 81 and 54% respectively on day 5. The lesion volumes of the two groups leveled out on day 7 (Fig. 3b).

**The immune-deficient mice did not exhibit similar differences in functional recoveries and lesion resolution as C57 mice**. The same surgical sequence as described above was performed on athymic nude and NOD-SCID mice. Interestingly, there was no difference in phenotypical results between the previous stroke groups and the control groups (Supplementary Fig. 1). The results indicated that adaptive immune components may account for the protective effect of a previous stroke on a subsequent stroke. As only T cells were absent in athymic nude, we hypothesized that T cells were likely to be participating in the observed alternative microglia polarization.

**Microglia demonstrated earlier alternative polarization in mice with a previous stroke, as demonstrated with flow cytometry and mRNA expression levels**. Flow cytometry revealed alternative chronological polarization profiles after the major hemorrhagic stroke (Fig. 4a). Both M1 and M2 levels returned to baseline in the sham and mini-stroke only groups before the major hemorrhagic stroke took place on day 0 (Fig. 4b, gray lines). M1 polarization, strictly defined as CD16/32$^+$CD206$^-$, saw a significant decline in the previous stroke group on day 5, while

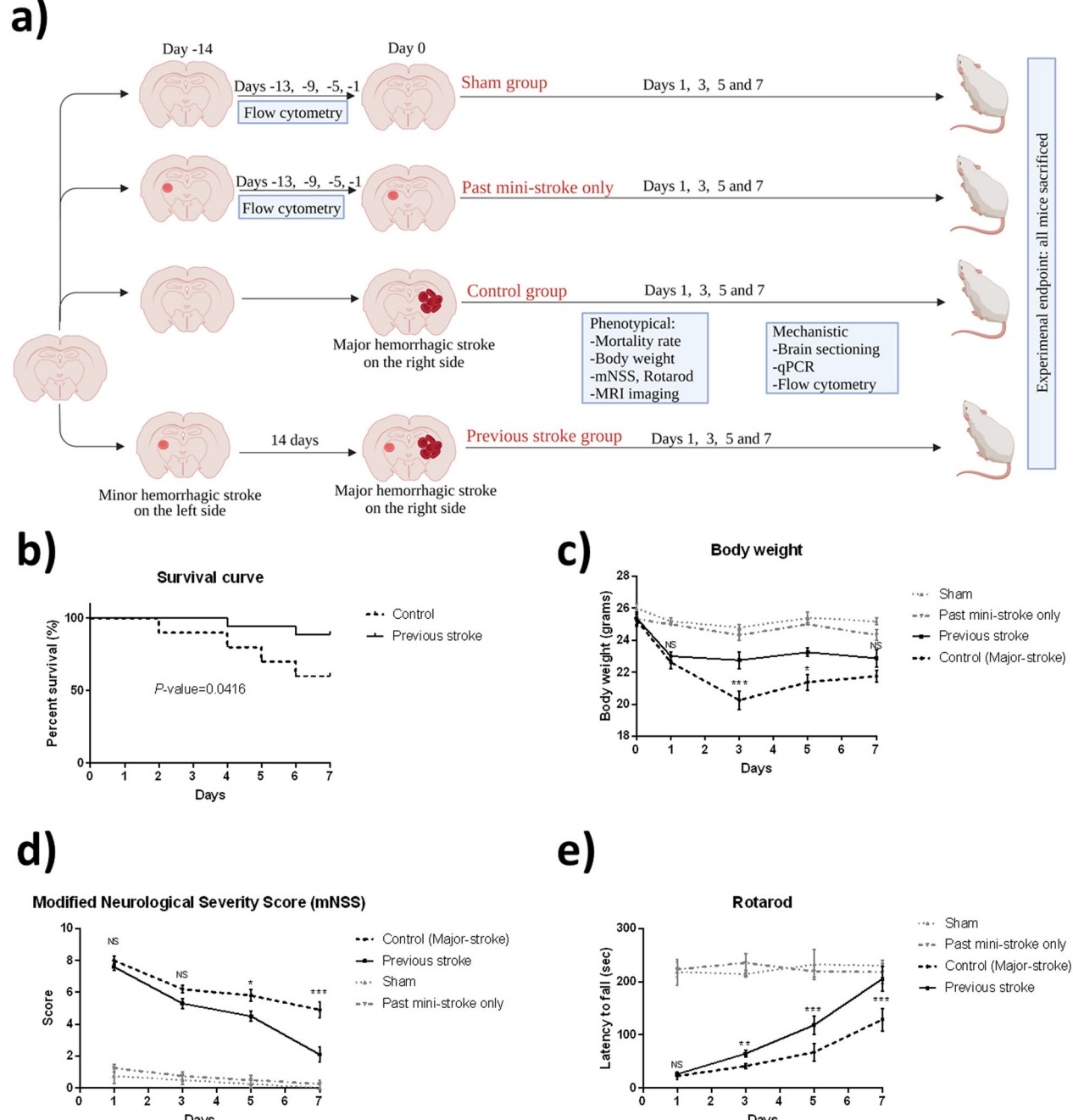

**Fig. 1 Functional outcome measurements after a hemorrhagic stroke. a** Sham group received two sham operations 14 days apart. Mice in the mini-stroke-only group received a minor stroke on day −14. Mice in the control group received only one major HS on the right, whereas those in the "previous stroke group" received a minor HS on the left 14 days prior to a major hemorrhagic stroke on the right. Flow cytometry was conducted on days −13, −9, −5, and −1. Phenotypical and mechanistic parameters were measured on days 1, 3, 5, and 7. Behavioral tests were conducted on days 1, 3, 5, and 7 after a major hemorrhagic stroke at the right basal ganglia. **b** Survival curve of the previous stroke group and control group ($n = 25$, Gehan–Breslow–Wilcoxon test, $p = 0.0416$). **c** Measurements of body weight ($n = 10$, RM two-way ANOVA, $p = 0.0183$). **d** Modified neurological severity score assessment ($n = 10$, RM two-way ANOVA, $p = 0.0003$). **e** Rotarod testing. Results presented as latency-to-fall (seconds) ($n = 10$, RM two-way ANOVA, < 0.0001). Error bars indicate SEM. *$p < 0.05$, **$p < 0.01$, NS not significant, mNSS modified Neurological Severity Score, MRI magnetic resonance imaging, qPCR quantitative polymerase chain reaction.

it remained high in the control group within the normal polarization plateau (Fig. 4b). The previous stroke group exhibited a sharp decline in M1 levels after an initial rise on day 3. Here, we observed the polarization pattern, as previous literature described[13,14], in the control group, with M1 expressional levels rising on day 3, plateauing on to day 5, and declining on day 7, to be replaced by an M2 predominant phenotype. However, within

the previous stroke group, this plateau was significantly lower in expressional levels, and the M1 dominant phase was cut short by a premature decline on day 5 (Fig. 4b). M2 polarization, defined as CD206$^+$CD16/32$^-$, rose in both groups as early as day 3. However, the previous stroke group did so to a much greater extent, surpassing the control group throughout days 3, 5, and 7. The expressional level of M2 in the control group correlated with

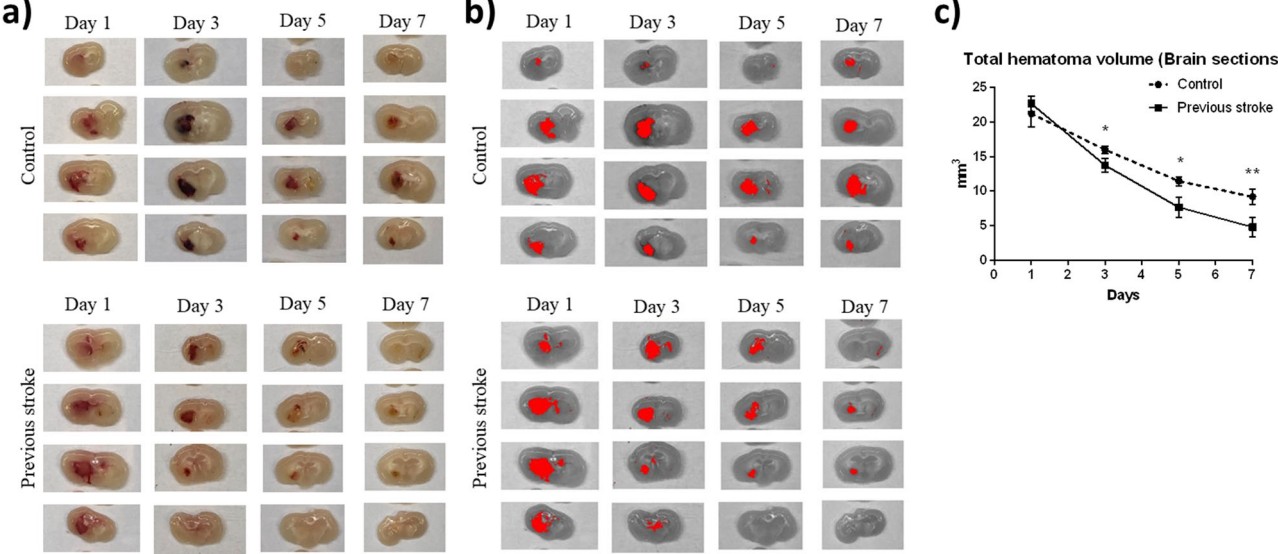

**Fig. 2 Coronal brain sections.** Mice were sacrificed on days 1, 3, 5, and 7, and perfused with 50 mL, 4 °C PBS. Brains were harvested and sliced into coronal sections of 2 mm. **a** Fresh coronal sections of control and previous group, in chronological order from left to right. **b** Images were imported to ImageJ software and transformed to 32-bit grayscale. Threshold gating was used to quantify the hematoma area of each slice. **c** Comparison of total hematoma volume between two groups. Results are expressed in mm$^3$. ($n = 5$ on each day) Error bars indicate SD. *$p < 0.05$, **$p < 0.01$, NS not significant.

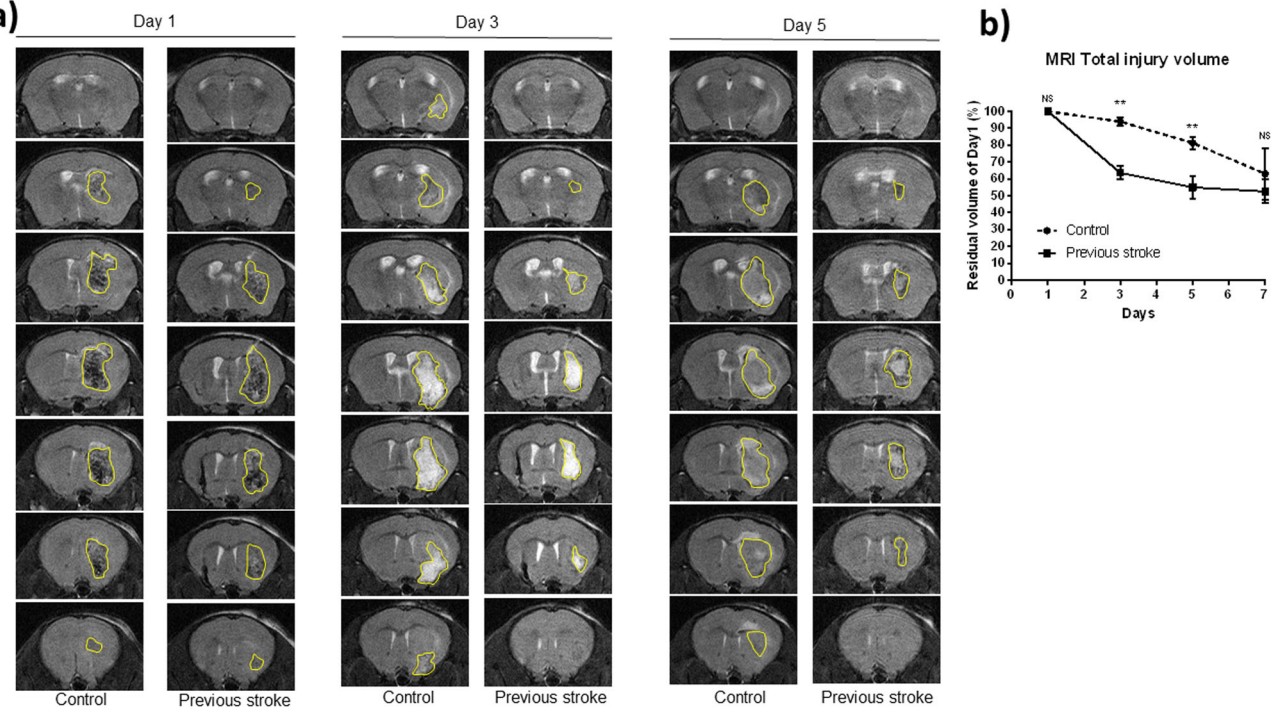

**Fig. 3 MRI images obtained on days 1, 3, and 5 after major hemorrhagic stroke induction. a** Coronal images, with maximal lesion area outlined in yellow, presented in chronological sequence. **b** Total lesion volume presented as a percentage of initial lesion volume on day 1 ($n = 5$ for each day). Error bars indicate SD. *$p < 0.05$, **$p < 0.01$, NS not significant.

that of past literature, gradually rising through days 3 and 5, and can be expected to surpass M1 after day 7. The previous stroke group, however, saw a significant increase on day 3 and remained at the same level throughout days 3, 5, and 7 without a dramatic fall within the period (Fig. 4b). M2 polarization was further verified with a double expression of CD206 and CD163, which correlated with the main results (Fig. 4c). In short, M1 markers were much attenuated in the previous stroke group compared to the control group, whereas M2 markers saw a more rapid increase

in the previous stroke group throughout days 3–5. qPCR analysis of in vivo brain tissue, although not statistically significant, also detected a more rapid rise in the M1 secreted, pro-inflammatory cytokine TNF-α within the previous stroke group, coupled with a faster decline in expression level on day 5, whereas the control group saw a latent yet much higher rise in TNF-α on day 5. Both groups saw a decrease in expression level on day 7 (Fig. 4d). iNOS, an M1 product indicative of inflammation, was found to be continuously more elevated in the control group throughout the

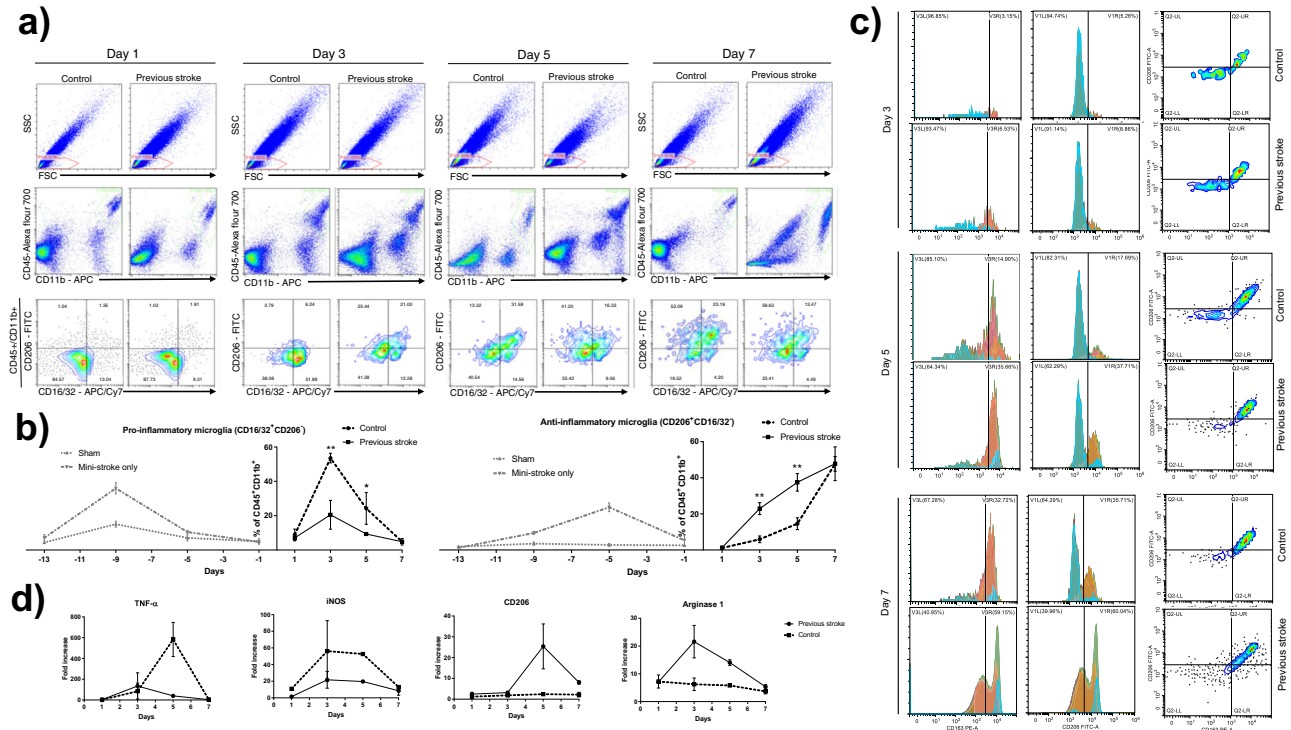

**Fig. 4 Flow cytometry and mRNA analysis of in vivo peri-hematomal tissue. a** Leukocytes were first gated from the FSC/SSC plot. Microglia were identified among leukocytes as $CD45^+CD11b^+$ cells. Significant elevation of $CD16/32^+CD206^-$ cells (M1 microglia) was seen in both groups on day 3, but the previous stroke group had a concomitant elevation in $CD206^+CD16/32^-$ cells, (M2 microglia) which the control group did not have. On day 5, both groups were presenting CD16/32 and CD206 markers, yet the percentage of $CD206^+CD16/32^-$ cells continued to be significantly higher in the previous stroke group, whereas $CD16/32^+CD206^-$ cells continued to be lower. **b** Left graph displaying the calculated percentage of $CD45^+CD11b^+$, $CD16/32^+CD206^-$ cells at each time point, representing the M1 microglia. Right graph displaying the calculated percentage of $CD45^+CD11b^+$, $CD206^+CD16/32^-$ cells at each time point, representing M2 microglia ($n = 7$ on each day). Both M1 and M2 levels returned to baseline levels in the sham and mini-stroke only groups before the major hemorrhagic stroke took place on day 0 (Gray lines; $n = 4$ on each day). Error bars indicate SEM. *$p < 0.05$ **$p < 0.01$. **c** Simultaneous expression of CD163 and CD206 revealed a near-identical percentage of double-positive cells each day, further verifying the M2 polarization profile. **d** qPCR analysis of in vivo samples. Mice were sacrificed on days 1, 3, 5, and 7. The right peri-hematomal region was obtained for mRNA extraction and analysis. TNF-α was elevated earlier in the previous stroke group but reached a much higher peak in the control group on day 5. iNOS was continuously more elevated in the control group. CD206 was elevated on day 5 in the previous stroke group but stayed low within the control group. Arginase-1 levels rose on day 3 within the previous stroke group but stayed low within the control group ($n = 4$ on each day). Error bars indicate SD. The results of the qPCR analysis were not statistically significant.

period. The two groups followed a similar elevation profile, to different extents, plateauing through days 3 and 5, and subsequently declining on day 7 (Fig. 4d). Arginase-1, a product of the M2 phenotype, was found to be highly elevated on days 3 and 5 in the previous stroke group, whereas its level remained low in the control group for the first 7 days. CD206, a surface marker for M2, was elevated in the previous stroke group on days 5 and 7, exceeding its counterpart, which remained low for the first 7 days (Fig. 4d). For illustrative purposes, immunofluorescence CD11b was used to stain microglia of coronal brain sections on day 3 post-HS; CD163 was used for M2 microglia. Brain sections at the perilesional area showed increased expression of M2 microglia marker in the previous stroke group compared to that of the control group (Supplementary Fig. 2).

**Microglia demonstrated earlier alternative polarization with a previous RBC coculture in vitro with the permissive effect of T cells.** Primary-cultured microglia that were separated mechanically from astrocytes were stained with Iba-1 (microglia surface marker) and DAPI (nuclear staining). Merged images demonstrated a >95% purity of microglia (Fig. 5a). Microglia were then divided into 6 groups, as illustrated in Fig. 5b. Based on our in vivo findings that immune-deficient mice exhibited

different results compared to C57 mice, we aimed to identify the role of T cells in the alteration of microglia activation with a previous stimulus in vitro. It has also been reported in past literature that T cells participate in the polarization of peripheral M2 macrophages[15]. Here we compared microglia that were exposed to RBCs at the second time point of day 14 (Groups D, E, and F). Groups A, B, and C served as baseline controls for the above and demonstrated that T cells alone (Group C) and a previous RBC exposure alone (Group B) did not have lasting effects on baseline microglia polarization 14 days later (Fig. 5c–f). Therefore, the results between Groups D, E, and F could be compared without bias.

At 24 h, the percentage of M1 phenotype ($CD16/32^+CD206^-$) drastically increased in Groups D, E, and F. Among them, both Groups E and F saw a more distinct early rise in M1 markers, compared to that of Group D (Fig. 5c). This indicated an earlier rise in M1 levels in the previously RBC-cocultured groups, regardless of the presence of T cells. M2 phenotype also increased in all three groups at 24 h after RBC coculture (Fig. 5d). Interestingly, only Group F (with T cells and a previous RBC coculture), and not Group E (previous RBC coculture only), had a significant increase in M2 phenotype compared to Group D (no previous RBC coculture) at 72 h (Fig. 5f). At 72 h, the M1 presentation rate dropped most significantly in Group F (with

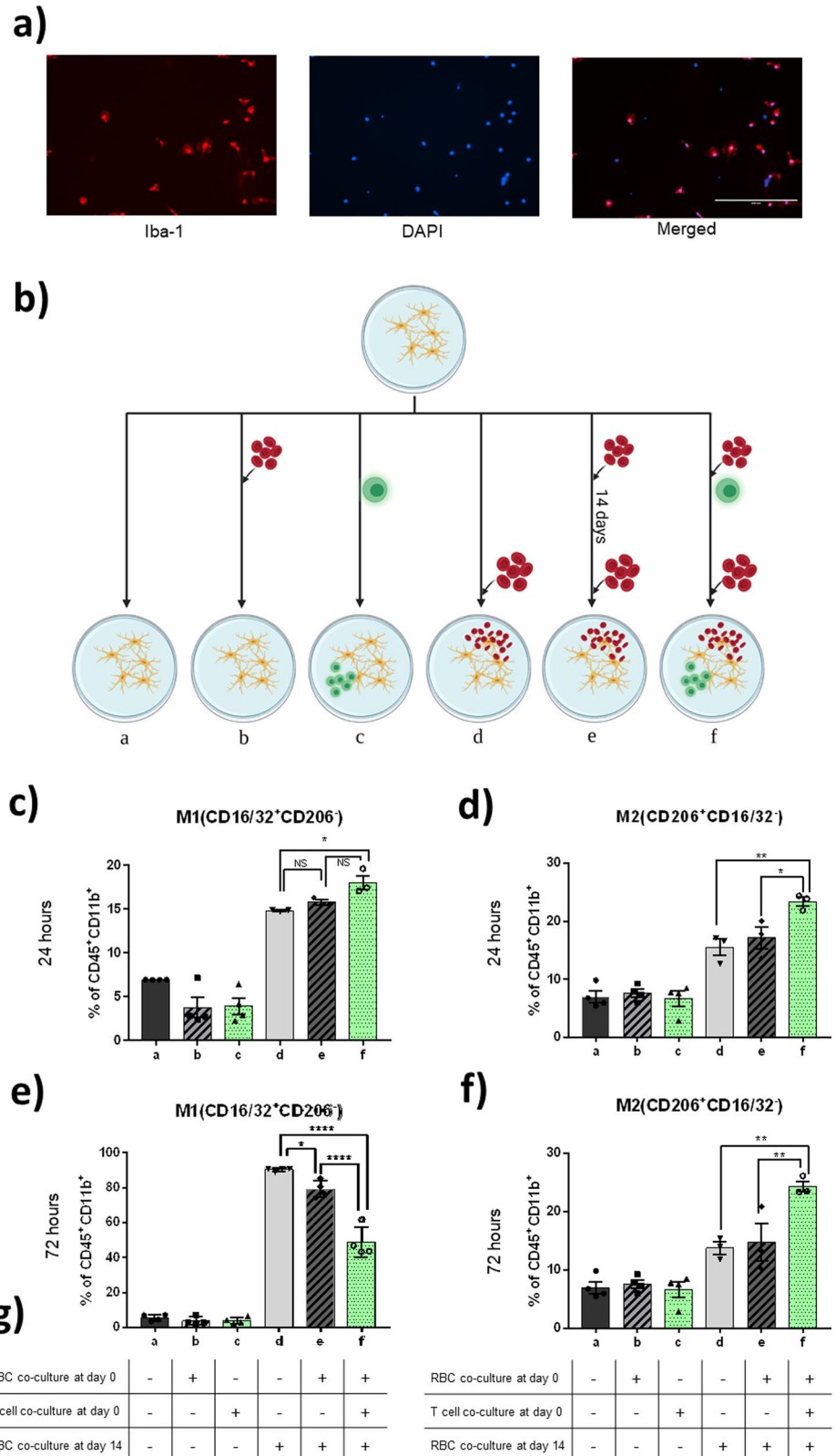

T cells), less so in Group E (without T cells), and remained high in Group D (without previous RBC coculture) (Fig. 5e). M2 presentation increased in all three groups, with the highest percentage seen in Group F, and no significant difference between Groups D and E (Fig. 5f). These results indicate that T cells had a permissive effect on altering the polarization profile of microglia following previous exposure to RBC.

During our erythrophagocytosis study, when fluorescent-labeled RBCs were added to microglia, the percentage of intracellular RBCs was highest in Group F (with T cells and a previous RBC coculture), second in Group E (with a previous RBC coculture) and lowest in Group D (RBC-naive microglia) (Fig. 6). This indicates superior RBC phagocytic abilities in microglia with a previous stimulus, which is enhanced by the presence of T cells.

**Fig. 5 In vitro experimental results. a** Microglia harvested from newborn pups were plated onto T75 flasks and separated from adherent astrocytes by mechanical detachment. Immunofluorescence staining revealed a successful primary culture of microglia with >95% purity. **b** Primary-cultured microglia were randomly divided into six groups. Group A received no stimulus. Group B received a single RBC coculture 14 days prior to analysis. Group C was cocultured with T cells. Group D had a single RBC coculture before detachment and analysis. Group E received two separate RBC cocultures. Group F received two separate RBC cocultures along with T cell coculture immediately after the first RBC coculture. Yellow cells: microglia, green cells: pan-naïve T cell, red cells: RBCs. **c–f** Flow cytometry analysis of in vitro primary-cultured microglia. Cells were collected at 24 and 72 h after RBC coculture. Bars marked in green represent the addition of pan-naïve T cells. **c** At 24 h post-RBC coculture, both groups with a previous RBC coculture (Groups E and F) presented with a more rapid increase in M1 polarization, compared to Group D, which is encountering RBCs for the first time. **d** At the same time, M2 expressional levels were higher in Groups E and F, compared to Group D. **e** At 72 h post-RBC coculture, Groups E and F (with or without a previous RBC coculture), had a significant difference in M1 expression levels. Surprisingly, when comparing Groups E and F (with or without T cells) there was also a significant difference, with a greater decrease in Group F. **f** Simultaneously, M2 markers were elevated in all three groups, but no difference between Groups D and E were seen, only when comparing them to Group F (with pan-naïve T cells) does significant difference appear. **g** Simple table reiterating the culture sequence as described in Methods. Error bars indicate SEM. *$p < 0.05$, **$p < 0.01$.

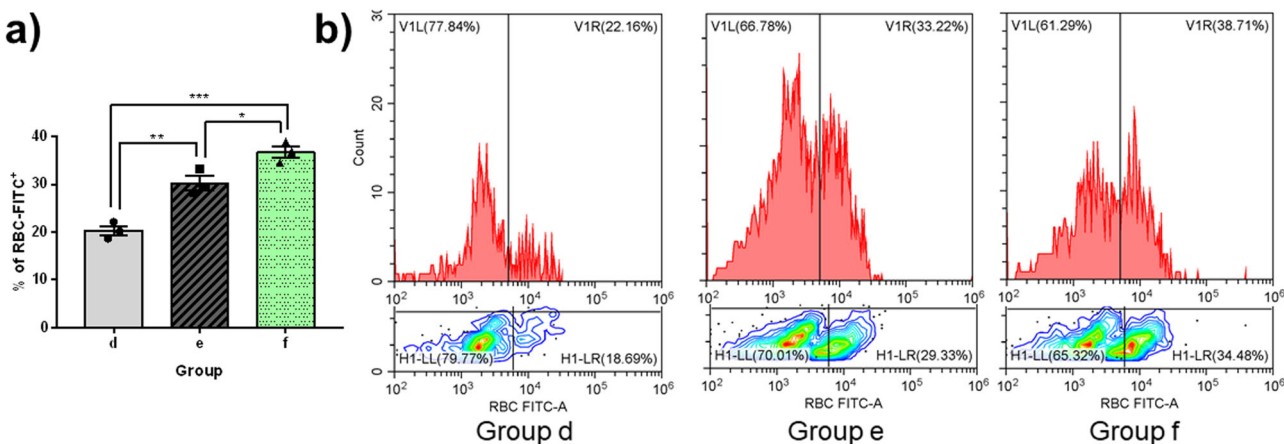

**Fig. 6 Erythrophagocytosis study on primary-cultured microglia in vitro.** PKH-26 labeled whole RBCs were cocultured with microglia for 1 h before being removed by PBS washing. Engulfed fluorescent RBCs remained within the flask-adherent microglia and were analyzed via flow cytometry with the FITC channel. **a** Phagocytic activity was highest in Group F (previous RBC coculture with T cells), second highest in Group E (previous RBC coculture), and lowest in Group D (no previous RBC coculture). Error bars indicate SEM. *$p < 0.05$, **$p < 0.01$. **b** Graphical representation of RBC-FITC$^+$ percentage.

## Discussion

In the clinical setting, the impact of a previous stroke on subsequent strokes has not been examined and remains unclear. The "protective" effect of a previous stroke on a subsequent stroke, a never before proposed concept, was demonstrated in a novelly designed, two-stroke procedural sequence in our study.

Our present work, to the best of our knowledge, is the first study to utilize a previous stroke as a form of immunological stimulus protection against subsequent strokes. Mice with a previous stroke exhibited better functional revival as soon as 3 days after a second stroke, coupled with increased survival rates and body weight improvement. Given the long-established clinical relationship between total lesion volume and performance status[16–18], a swift recovery could be credited to a faster resolution in hematoma volume and perilesional edema, as documented by our MRI imaging and histopathological brain sections. It is worthy to note that T2-weighted MRI imaging showed that perilesional edema starting from day 3 (as a hyperintense signal) occurred at a much higher volume in the control group. Those with a previous stroke exhibited substantially less perilesional edema, demonstrated by a smaller volume of peri-hematomal hyperintense signals throughout days 3–5. There has also been robust evidence showing that secondary inflammatory damage is associated with microglia recruitment and activation towards M1 at the perilesional area[19] whereas M2 would promote phagocytosis, neuronal regeneration, and attenuate inflammation[9]. Therefore, the increased rate of lesion resolution following a previous stroke led us to the hypothesis that microglia could be the main mediator of this protective effect.

Both microglia and bone marrow-derived macrophages may contribute to the immune sequelae of hemorrhagic stroke. Several studies have thus umbrella-termed intracerebral CD11b+ as microglia/peripherally-derived monocytes[12,20]. We, however, adopted a more narrowly defined working hypothesis that only looks at microglia in vitro. Since the cells that were primary-cultured from newborn pups were microglia-only, the fact that they demonstrated preferential M2 polarization and increased erythrophagocytosis after a simulated mini-stroke enabled us to ascertain with reasonable certainty and specificity that microglia alone were at least partially responsible for eliciting the observed response.

Microglia, acting as the main immune surveillance within the brain, are the first responders to the site of hemorrhage and contributors to lesion resolution. Classically activated microglia demonstrate pro-inflammatory characteristics and are labeled as "M1"-like phenotype, whereas alternatively activated ones were generalized as "M2"-like phenotype, comprising of M2a, 2b, and 2c subclasses, promoting cell regeneration, erythrophagocytosis, hemoglobin, and cell debris clearance. Although the generalized term "M2" is not perfect, it adequately represents the spectrum of alternatively polarized cells we are observing. The dichotomy of activated microglia into M1 and M2 is admittedly an over-simplification, upon which, future research should be conducted to unpack the precise subtypes and their respective roles. As early as 3 h post-stroke, microglia would polarize to an M1 predominant phenotype, exerting inflammatory actions and further contributing to brain injury by TNF-α, iNOS, and CD16/32. Past

literature regarding HS has shown that M1 action peaks on day 3 before gradually declining to give rise to an M2 phenotype on day 7[9]. M2 microglia, contrary to M1, exerts anti-inflammatory actions and promotes cell regeneration, phagocytosis, and removal of cell debris. M2 spectrum microglia carries out actions via receptors such as CD206, CD163, CD36, and cytokines such as Arginase-1 and TGF-β[12,21]. In our study, the chronological polarization profile of the control group corresponded to that of past literature. However, those with a previous stroke had a premature decline in M1 and shifted towards M2 markers as early as days 3 and 5. This phenotypical switch, characterized in other studies as the end of secondary inflammatory damage and the commencement of tissue repair and debris clearance[13,14,22], occurred markedly earlier in mice with a previous HS. Microglia, like macrophages, were previously classified as part of the innate immune system, unable to facilitate immune specificity nor retain immune memory. This concept has been challenged in recent years by a novel study[10]. They demonstrated that multiple episodes of peripheral lipopolysaccharide stimulation led to either exacerbation or attenuation of pro-inflammatory (IL-1B) responses, focusing not on the 'adaptive' players, but rather on the "innate" ones. The significance of these findings is that innate immune cells could retain long-term memory with influence on neurological diseases[10]. The same concept has been proposed in macrophages in a septic shock model, in which macrophages showed immune tolerance following a previous septic shock[23]. In these studies, whether a previous stimulus would induce inflammatory training (exacerbation) or tolerance (attenuation) among immune cells depends on the dosage, duration, and type of the first stimulation[24]. The dichotomy of training and tolerance may be overgeneralized, however. In our study, for example, the phagocytic actions of M2 would be classified as training[24], and yet their anti-inflammatory actions would be classified as tolerance. To the best of our knowledge, there have not been any studies regarding innate immune memory that investigate the differences in M1/M2 polarization profile.

During stimulation, cells of the innate immune system (NK, Neutrophils, macrophages, and microglia) react rapidly via pattern response receptors (PRRs)[25]. PRRs respond to a wide array of microorganisms or non-microbial Danger-associated molecular patterns (DAMPs) with partial specificity[26]. These stimulations activate both signaling cascades intracellularly and facilitate cell-cell interactions. The former may include epigenetic changes such as acetylation or methylation via PRR stimulation pathways[27], causing downstream production of cytokines and chemokines, whereas the latter may involve TLR-PRR receptor crosstalk with T lymphocytes[25]. In macrophages, histone modifications such as H3K9me3 have been shown to prevent promotor activation and thus lead to immune tolerance[28], whereas H3K4me1 exacerbates inflammatory response and thus results in immune training[29]. These manifestations of innate immune memory in macrophages, which had many similarities to microglia, strengthened the evidence behind our observed phenomenon that microglia could indeed retain immune memory of a previous HS based on epigenetic transcriptome alterations. We surmise that a previous HS may induce a certain form of protective immune memory which had a lasting effect that remained in the background, only to be recalled when encountering a subsequent stroke.

Past literature has demonstrated that peripherally-derived T cells interact with microglia during an HS[30,31]. T cells egress peripheral lymph nodes via systemic signaling of Sphingosine 1-phosphate, and cross the blood-brain barrier (BBB) via VCAM-1 interaction[32]. Once within the brain, T cells are approached by microglia and receive signals via IL-6, 12, 23, or IL-10. T cells then send feedback signals to microglia via IFN-gamma, TNF,

and CSF-2, modulating microglia response and actions[32]. T cells also interact with microglia via a variety of membrane receptors, such as PDL1-PD1, CD40-CD40L, MHC2-TCR, and CD80/86-CD28[32]. Our negative results obtained from athymic nude mice, along with the literary evidence of T cell-microglia interaction, further confirmed our suspicion that the protective effect was T cell-related.

Experiments exposing microglia to RBCs were designed to mimic an HS in vitro in this study. This was achieved by adding whole RBCs into a microglia primary culture, as described by past literature[33]. This particular design aims to reproduce the primary insult that occurs within the brain: copious amounts of toxigenic RBCs extravasation across the compromised BBB during vessel rupture. By replicating this primary injury in a cell-based model, we only saw a modest difference between microglia that had received a previous insult compared with microglia that were encountering RBCs for the first time at 24 h. However, with the addition of pan-naïve T cells into the culture system immediately after the first RBC coculture, polarization changed drastically in the previously RBC-cocultured group at 24 and 72 h after a second RBC coculture. The T cells that were employed here were harvested from adult C57 mice, and only those without prior memory (CD45RO⁻) were selected, thus referred to as pan-naïve T cells, carrying no memory over from the mouse it was harvested from. As was observed, the presence of T cells after the initial RBC coculturing event was required to induce a significant difference between "primed" and "unprimed" microglia. During the first stimulus, the classically, then alternatively activated microglia, could have interacted with naïve T cells. This interaction may then induce innate memory formation, which resulted in alternative microglia polarization and increased erythrophagocytosis during a subsequent stroke.

Our findings have the potential to change the face of translational stroke treatment. The human correlate of a mini-stroke can take many forms, such as the introduction of small amounts of whole RBCs into the CSF via lumbar puncture which then induces dormant microglia memory within the central nervous system as a form of active immunity. Another approach is through passive immunity in the form of ex vivo cell therapy, such as introducing HLA-matched, trained microglia (or T cells) into the CSF within the early hours after a hemorrhagic stroke has occurred. Lastly, if the antigen presented on microglia could be identified and sequenced via proteomics, perhaps by combining it with microglia MHC-II, the end product would be able to induce immune memory formation to combat an upcoming, major hemorrhagic stroke or possibly other forms of intracranial hemorrhage such as traumatic hematomas. The overarching approach for future studies is to exploit the brain's innate immune response to a blood-specific trigger, introduced in a clinically benign way, so as to enhance the body's capabilities to deal with an actual clinical event. Suppose a clinically acceptable way to mimic a priming mini-stroke were to be developed. In that case, individuals at risk could receive the priming event as a prophylactic "first-hit", or broadly speaking, "vaccine-like" protection, should they develop a massive spontaneous hemorrhagic stroke in the future. No longer would there be a strict therapeutic time window, but rather, the treatment given beforehand would have a long-lasting effect on M2 preferential polarization.

However, the molecular mechanism underlying microglia differentiation had not been clearly investigated in our study. Multiple components of RBCs could have accounted for M2 polarization and thus the formation of microglia immune memory, including RBC membrane antigens, hemoglobin, thrombin, or ferrous iron. This could be investigated in the future to realize translational implementation. qPCR analysis was performed on whole-brain homogenates, which includes not only

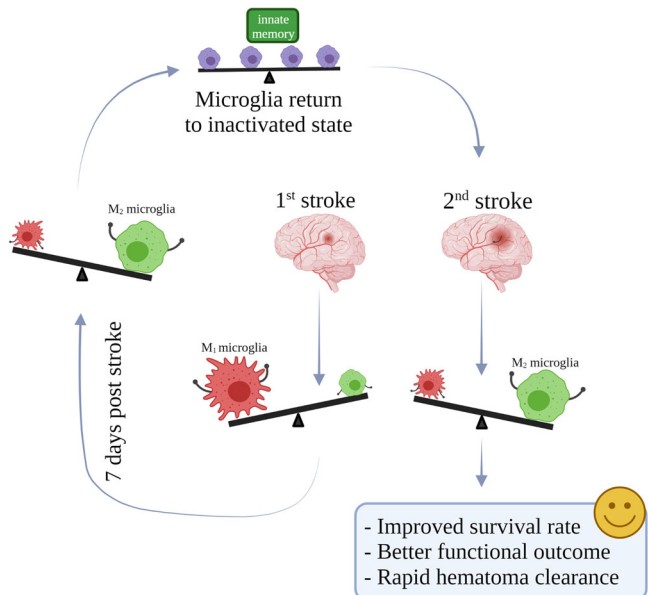

**Fig. 7 Graphic summary.** The first hemorrhagic stroke initiated a cascade of inflammatory actions downstream that corresponds to what others have observed in the past: M1 microglia take dominance initially before gradually declining to give way to an M2 predominant profile at day 7 post hemorrhagic stroke. Microglia then return to baseline levels. When a second hemorrhagic stroke was introduced, microglia no longer followed the former polarization profile but rather assumed an M2 predominant phenotype at an early stage. The group of mice with a previous stroke demonstrated improved survival rates, better functional outcomes, and rapid lesion resolution, compared to mice that were only encountering a hemorrhagic stroke for the first time.

microglia but also a mixture of astrocytes, neurons, necrotic tissue, and cell debris. Unlike flow cytometry, qPCR results did not show statistical significance. This could possibly be attributed to the wide range of interference from the aforementioned components. The exact role of T cells on microglia preferential M2 polarization in the second stroke was not fully discussed either, as was the mechanism of innate memory preservation of microglia. Also, the effects of a previous stroke on subsequent strokes may involve other participants of the immune system, such as neutrophils, B cells, and NK cells[34]. The role of axon and gray matter degeneration could also be of importance, as they crosstalk with microglia after HS[35]. Astrocytes and oligodendrocytes also secrete chemokines such as CCL2, CXCL1, and CXCL10, which might have roles in microglial differentiation and could be explored in the future studies[9]. Once we can identify the exact priming antigen and means of conferring such innate memory, a potential therapeutic target will thus be born and avenue to modify microglia M1/2 response either in the acute phase of patients who had HS or as prophylaxis in those at high risk of developing one. Future studies could also aim to verify the findings in the clinical setting by prospectively examining hematoma resorption and functional recovery in patients with recurrent HS.

This study provided evidence on a newly discovered trend that had never been discussed in past literature. The protective effects of a previous hemorrhagic stroke on subsequent strokes were demonstrated via improved performance in survival, behavioral, and MRI imaging. Microglia polarization was shown to skew towards an alternatively polarized spectrum at an earlier phase in mice with a previous hemorrhagic stroke (Fig. 7). In vitro studies revealed that microglia had retained innate immune memory during the previous stroke and that T cells were involved in this

protective mechanism regarding earlier alternative microglia polarization. Our incidental discovery presented new evidence on a mechanism regarding the potential clinical impacts of successive hemorrhagic strokes which may pave the road to new, anticipatory therapeutic interventions that have never been seen before.

## Methods

**Hemorrhagic stroke animal model.** All animal experiments were approved by the Committee on the Use of Live Animals in Teaching and Research of our institution. Male C57 (C57BL/6 N, Charles River Lab, USA) mice 10–12 weeks old underwent all the described experiments unless otherwise specified (e.g., immune-deficient mice experiment design).

Mice were randomly divided into four groups. The sham group underwent two sham operations on days −14 and 0. The "past mini-stroke only" group underwent a single minor stroke on day −14. The "previous stroke group" underwent a minor HS at the left basal ganglia on day 14 and a major HS on the right on day 0. The control group received only one major HS at the right basal ganglia, with a sham operation on day −14. (Fig. 1a). For induction of the major HS, a replicable model was conducted as described before[13]. Mice were anesthetized via intraperitoneal injection of ketamine (100 mg/kg) and xylazine (10 mg/kg). A stereotactic device was used. Mice were secured in a prone position, and scalp fur was removed. A midline scalp single incision was made. Intracranial bleeding was induced by inserting a 26-gauge needle (10 uL Hamilton syringe) at 3.5 mm below the dura into the right basal ganglia at coordinates (x, y) = (+0.2 mm, +0.05 mm), with the bregma as (0,0), through a burr hole. One minute later, 0.1 units of collagenase (0.25 U/μL, 0.4 μL) (Sigma-Aldrich, St. Louis, MI, United States) was injected at a rate of 0.1 μL/min with an infusion pump. After needle retraction, the burr hole was covered with bone wax, and the scalp was sutured. For a minor HS induction, the procedure was the same as mentioned above, with the only difference being that collagenase dosage was adjusted to 0.04 U (0.2 U/μL, 0.2 μL), and the injection site was at the left basal ganglia (−0.02 mm, +0.5 mm). Replicable hematomas were induced using this dosage and method for the minor HS (Supplementary Fig. 3). For a sham operation, the mice had a bur hole drilled at the same coordinates with needle insertion for 10 min before retraction.

In order to observe whether an absence of certain immune cells could interfere with functional recovery under the above study design, we used immune-modified mouse strains. Athymic nude mice (BALB/cAnN-nu, Charles River Lab, USA) and NOD-SCID mice (NOD.CB17-Prkdcscid/J, The Jackson Laboratory, USA) were used to repeat phenotypical experiments—similar sequences of HS for the "previous stroke group" and control group were performed on athymic nude mice, which lack mature T cells, and NOD-SCID mice, which lack mature B cells, T cell, and have defective NK cells.

**Phenotypical observations.** Mice were monitored on days 1, 3, 5, and 7 following the induction of a major hemorrhagic stroke. A blinded observer took the measurements.

**Modified neurological severity score (mNSS).** The mNSS test, first described by Chen et al.[36] and verified by validation studies[37], quantifies motor, sensory, and balance deficits post-stroke. With a maximum score of 18 being the most severe, the gradual decline in neurological deficit scores traces the speed and extent of motor and sensory recovery after the initial injury[37].

**Rotarod.** The Rotarod apparatus (AccuRotor, Omnitech electronics) rotates at an acceleration of 0.133 cm/s[2]. Results are recorded as latency-to-fall, which is the duration between commencement of rotation until the mouse falls off the rod[36]. Mice were first trained 7 days in three separate sessions prior to the major HS event. Subsequently, mice were assessed by the Rotarod test with the same acceleration profile. For each mouse, three runs were repeated, with 20-min rest intervals in-between.

**Body weight.** Mice were weighed individually every morning on days 1, 3, 5, and 7. Results were documented in grams.

**Histopathologic examination.** Mice were sacrificed on days 1, 3, 5, and 7 and perfused with 50 mL 4 °C normal saline. For the mini-stroke only and sham groups, mice were sacrificed 13, 9, 5, and 1 day before the major hemorrhagic stroke of other groups took place. Brains were sectioned into coronal slices of 2 mm thickness, and a photo was taken of each coronal section. Coronal images were imported to ImageJ, converted to grayscale, and a gradient-gated threshold was used to obtain the hematoma area. The sum of the sections from the cranial-caudal and caudal-cranial views were obtained, and the mean was multiplied by 2 mm (slice thickness) to obtain a final value of hematoma volume (expressed in mm[3]).

**Magnetic resonance imaging (MRI) studies**. A 7 T MRI scanner (PharmaScan 70/16, Bruder Biospin GmbH) was used to take 12 coronal plain scans, each of 0.5 mm thickness and positioned perpendicular to the axial and sagittal axes as previously described[38]. T2-weighted images were acquired as anatomical reference using a Rapid Acquisition with Refocused Echoes (RARE) sequence (FOV = 32 × 32 mm², data matrix = 256 × 256, RARE factor = 8, TE/TR = 36/4200 ms). Images of the coronal plain were imported into ImageJ (version 1.8.0 for Windows, ImageJ Software). For each slice of the 12 coronal images, the total lesion area, defined as the hyperintense core, periphery, and surrounding rim[39] was outlined manually. The sum of the combined lesion area from the 12 coronal slices was multiplied by slice thickness (0.5 mm) to obtain the total volume of the lesion. For statistical analysis, the total lesion volume was compared chronologically within the same mouse, with that on day 1 being presented as 100%. A subsequent decline in injury volume is expressed as a decrease in the percentage of residual volume compared to that on day 1.

**Quantitative polymerase chain reaction (qPCR)**. Mice were sacrificed 1, 3, 5, and 7 days after HS and perfused with 50 mL 4 °C normal saline. Brains were sectioned into slices of 2 mm thickness. Starting from the olfactory bulbs, the fourth and fifth slices were those with the largest hematoma area. These slices were harvested and subsequently divided into left and right sections. The right (injured) sections and enzymatically digested with RIPA buffer and PI. RNA was extracted from digested tissue, and cDNA was synthesized using a commercially available kit (Takara PrimeScript RT Reagent Kit). qRT-PCR using SYBR Green (QIAGEN) was performed to detect the subsequent genes: TNF-a, iNOS, CD206, and Arginine-1. The threshold cycle (CT) was normalized to GAPDH (ΔCT) of the same mouse and then to that of the sham mouse (ΔΔCT). Mean fold changes are depicted as $2^{(-\Delta\Delta CT)}$. Data acquisition was conducted in technical triplets. Primers are presented in Table 1.

**Immunofluorescence staining**. Mice were sacrificed and perfused with 50 mL cold PBS, followed by 50 mL 4% paraformaldehyde. Cryosections were obtained of the coronal plain at 5 µm thickness. Non-specific staining was blocked by 1% BSA for 1 h, followed by incubation with primary antibodies against microglia surface marker CD11b (1:100 Cell Signaling, Danvers, MA, USA), M2 marker CD163 (1:120 Cell Signaling) for 2 h at 4 °C. Slides were washed thrice before staining with secondary antibodies: anti-rabbit (Cell Signaling) and anti-mouse (Cell Signaling). DAPI nuclear staining (Sigma-Aldrich) was added to the mounting solution. Primary-cultured microglia were stained as follows: cells were fixed with 4% paraformaldehyde for 1 h, washed thrice with cold PBS, and stained with antibodies against Iba-1(1:100, Cell Signaling) for 2 h with nuclear staining DAPI (Sigma-Aldrich) added at the last 15 min. Cells were washed twice before incubation with secondary antibodies: anti-mouse (Cell Signaling) and anti-rabbit (Cell Signaling). All images were taken by the Invitrogen EVOS FL Color imaging system.

**Flow cytometry**. Mice from the previous stroke group and control group were sacrificed on days 1, 3, 5, and 7. Mice from the sham group and "past mini-stroke only" group were sacrificed on days −13, −9, −5, and −1. Peri-hematomal brain tissue was collected and mechanically homogenized by cutting with scissors, then digested by 5 mL 4% collagenase (Sigma-Aldrich, St. Louis, MI, United States) for 15 min at 37 °C. Digestion was halted by adding a 5 mL culture medium and single-cell suspension obtained by filtering through a 75 µm cell strainer. Cells were centrifuged at 400 × g for 5 min and resuspended in PBS. Cells were then washed again and resuspended in Flow Cytometry buffer (Cell signaling, Danvers, MA, USA) at a concentration of 10⁵ cells per mL. For in vitro studies, microglia were scraped from the flask by a cell scraper at the experimental endpoint of each group. Cells were washed twice by centrifuging at 400 × g for 5 min and suspended in flow cytometry buffer. Each mL of suspended cells was incubated with conjugated flow antibodies (APC/Cy7 anti-mouse CD16/32 antibody, Biolegend #101327, 1:100;

FITC anti-mouse CD206 antibody, Biolegend #141703, 1:80; Alexa Fluor 700 anti-mouse CD45.2 antibody, Biolegend #109822, 1:70; APC anti-mouse/human CD11b, Biolegend #101212, 1:100) for 1 h in darkness. For M2 verification, cells were additionally stained with PE anti-mouse CD163 antibody, Biolegend #156703. Cells were then washed twice by centrifuging at 400 × g for 5 min and resuspended in Flow Cytometry buffer. Analysis was conducted by CytoFLEX Flow Cytometer (Beckman Life Sciences, Indianapolis, IN, USA).

**Microglia isolation and cell culture**. In vitro experiments were conducted with primary-cultured microglia harvested from newborn pups of 0–3 days old, as previously described[40]. Pups were sacrificed by intraperitoneal injection of phenobarbital, then sprayed with 70% alcohol and decapitated with sterile equipment. The decapitated heads were rinsed in 4 °C PBS and immersed in dissection media. Under a microscope, the scalp was removed and the skull dissected along the midline up to the bregma, so that the brain could be scooped out using curving forceps. 5–6 brains were pooled and mechanically homogenized by cutting with scissors, then digested with 0.25% trypsin at 37 °C for 15 min. To end the digestion process, a 1.2 ml 1 mg/ml trypsin inhibitor was added and incubated for 1 min. The homogenized products were then centrifuged at 400 × g for 5 min and resuspended in warmed culture media (90% Dulbecco's modified Eagle medium, Thermo Fisher Scientific, Gibco™; 10% fetal bovine serum, GE Healthcare, Hyclone™) twice, before being plated onto PDL coated T75 flasks at a density of 50,000 cells/cm². Culture media was changed the next day, and subsequently every 5 days. This first step yields a confluent layer of astrocytes and microglia after 7 days.

To separate microglia from strongly adherent astrocytes, the flasks were shaken at 200 rpm for 2 h in a cell shaker maintained at 37 °C. The floating cells were collected and transferred to a newly coated T75 flask. This process yielded high quantity and high-quality microglia, which are ready for experimental use the next day.

**Red blood cell (RBC) coculture**. In order to mimic an HS in the cell culture setting, we added whole RBCs collected from the tail vein of mice at a ratio of 10:1 (RBC to microglia), as described before[41]. The coculture system was maintained at 37 °C for 1 h, and RBCs were subsequently removed via changing of cell medium. RBCs were non-adherent, while microglia were adherent to the flask wall.

**Isolation of T cells**. Splenic cells were obtained from adult C57 mice by homogenizing the whole spleen mechanically with scissors. RBCs were removed via lysis medium (Thermo Fisher, Waltham, MA, USA). Pan-naïve T cells were extracted via column-free magnetic cell isolation: first by positive selection (CD3 + CD45RA + CD197 + ), then negative selection (CD45RO−) using magnetic beads and a commercial kit (EasySep™ Human Naïve Pan T Cell Isolation Kit).

The collected pan-naïve T cells were added to Group F immediately after the first RBC coculture event (without direct contact with the RBCs) at a ratio of 1:1 (microglia to T cell). T cells were added to group C at the same time.

**Erythrophagocytosis study**. Whole RBCs were labeled with PKH-26 fluorescent probe (Sigma-Aldrich) and cocultured with primary microglia at a ratio of 10:1, as previously described[33]. After incubation at 37 °C for 1 h, the unengulfed RBCs were removed by washing thrice with PBS. Microglia were then harvested from the flask wall. The engulfed, intra-microglial RBCs were detected by flow cytometry via the FITC channel. The level of phagocytosis was quantified by the positive FITC percentage of each measurement.

**Statistics and reproducibility**. Phenotypical studies that include multiple measurements of the same mouse over time (body weight, mNSS, Rotarod, MRI total lesion volume) were analyzed by repeated-measurements two-way ANOVA with matched values of the same mouse across time ($n = 10$ for the previous stroke and control groups, $n = 5$ for the mini-stroke only and sham groups). Additionally, Sidak's multiple comparisons test were conducted on the mean value obtained on each date between experimental groups ($n = 25$). Mechanistic studies that require sacrificing mice at the time of measurement (flow cytometry, qPCR) were analyzed by regular two-way ANOVA (no repeated measures) and Sidak's multiple comparisons test ($n = 7$ for the previous stroke and control groups; $n = 4$ for the mini-stroke only and sham groups). Cell line studies involving more than two groups of importance to compare were analyzed by one-way ANOVA with Tukey's multiple comparisons test. Cell line experiments were repeated three times within each group. The Gehan–Breslow–Wilcoxon test was used in the Kaplan–Meier survival curve analysis. A $p$ value < 0.05 was considered statistically significant. GraphPad Prism version 6.04 for Windows, GraphPad Software was used.

**Table 1 Primer sequences for the qPCR analysis conducted on in vivo brain section samples on days 1, 3, 5, and 7 after a major hemorrhagic stroke.**

| | |
|---|---|
| miNOS - F | 5′GCTTGTCTCTGGGTCCTCTG3′ |
| miNOS - R | 5′CTCACTGGGACAGCACAGAA3′ |
| mCD206 - F | 5′AACAAGAATGGTGGGCAGTC3′ |
| mCD206 - R | 5′CCTTTCAGTCCTTTGCAAGC3′ |
| mTNFα - F | 5′GCCTCCCTCTCAGTTCT3′ |
| mTNFα - R | 5′ACTTGGTGGTTTGCTACGAC3′ |
| mArginase 1 - F | 5′GCTTGCTTCGGAACTCAAC3′ |
| mArginase 1 - R | 5′CGCATTCACAGTCACTTAGG3′ |
| mGAPDH - F | 5′GCCAAGGCTGTGGGCAAGGT3′ |
| mGAPDH - R | 5′TCTCCAGGCGGCACGTCAGA3′ |

**Reporting summary**. Further information on research design is available in the Nature Research Reporting Summary linked to this article.

## Data availability
Source data supporting the figures of this study is included in Supplementary Data 1.

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

## Acknowledgements
We would like to thank Professor Wu, Ed Xuekui, and his lab for MRI machine usage. We would also like to acknowledge the technical support of the Animal Laboratory Unit of our department. Figures 1a, 5b, and 7 were created with BioRender.com.

## Author contributions
S.-S.L.: Conceptualized, designed, performed the research, and drafted the manuscript. L.P.: Designed, supervised the research, and critically revised the manuscript. Y.C.: Designed, supervised the research, and reviewed the manuscript. J.X.L. and A.C.K.N. reviewed the manuscript. G.K.K.L.: Conceptualized, designed, supervised the research, and critically revised the manuscript.

## Competing interests
The authors declare no competing interests.

## Ethical approval
All animal experiments were approved by the Committee on the Use of Live Animals in Teaching and Research of our institution.
