## [Peer Review File · Communications Biology]

Reviewers' comments:

Reviewer #1 (Remarks to the Author):

This report is to document that a previous 'mini' ICH served as a protective role in a subsequent ICH, through enhancing T cell-dependent M2 microglia polarisation. This is an interesting topic.

There are some suggestions for the research team:

1. Which sequence of MRI was used to evaluate the lesion? It should be described in the section of Method and materials.
2. As the authors stated, the lesion volume includes hematoma, edema, and surrounding rim, so the volume reduction does not represent the hematoma clearance.
3. The author attempted to prove the protective effect of previous 'mini'stroke is through enhancing M2 microglia polarisation. So the information of M2 microglia polarization and T cell activation after the first ICH is important.
4. It is better to provide two positive biomarkers to confirm microglia polarization.
5. Weather small-dose collagenase could induce mini-ICH? Is the volume of hematoma stable in the first ICH? It is better to use the autologous blood injection model to control the hematoma volume.

Reviewer #2 (Remarks to the Author):

The study performed by Lee et al demonstrated that a past 'mini-stroke resulted in better neurological function recovery, faster attenuation of hematoma and lesion volume, and better survival of hemorrhagic stroke mice, in a T cell-dependent manner. The study is interesting and could provide a therapeutic mechanism of hemorrhagic stroke.

1. The authors used major hemorrhagic stroke as a control group. However, I think they need to add a mini-stroke-only group and sham group in the study, to test if mini-stroke-only has any effects on brain injury and recovery.
2. In this study, M1 and M2 microglia are over-simplistic concepts of microglia. I would suggest not to use M1 and M2, but using pro-inflammatory microglia and anti-inflammatory microglia.
3. CD16/32, iNOS, CD206, Arginase are not specific markers of M1 and M2 microglia. I would suggest to use double immunostaining with IBA-1 to evaluate microglia polarization
4. Besides microglia, macrophages will infiltrate into the brain after hemorrhagic stroke. What are the roles that macrophages played in mini-stroke induced protective effects?

Reviewer #3 (Remarks to the Author):

In the manuscript entitled „A previous hemorrhagic stroke protects against a subsequent stroke via microglia alternative polarisation“ the authors Lee et al. describe the interesting and novel observation that the modulation of microglial polarization via a previous hemorrhage can attenuate neuronal injury in a murine mouse model of hemorrhagic cerebral stroke.

Although the authors present some interesting results, I would like to raise a couple of points for improvement, especially concerning the complex cellular interaction within the brain and translational applicability. As a general opinion, the presented data present an excellent basis to report the interesting observation, but omit some important aspects that should be addressed:

1. The authors state a difference in survival depending on the pre-stroke intervention (Figure 1B). Was this difference statistically significant? This information is missing from the text or the figure legend.
2. The authors make the main argument that the modification of microglial polarization leads to improved neurological outcome due to faster hematoma clearance (Figure 2/3). But what is the mechanism of reduced hematoma size? Is it increased erythrophagocytosis and cell debris removal by

microglia? Is it reduction of brain edema and blood-brain barrier disruption? Is it really faster hematoma removal rather than less susceptibility to continued bleeding ("chronic" collagenase bleeding model, vascular stability)? Is it neuronal regeneration? Mechanistical studies apart from inflammatory cytokine production are necessary to proof that microglial polarization (which the authors clearly show) is causal for the protective effects and not a consequence of unrelated changes in the microenvironment of the brain. This could be established by either investigating the above mentioned mechanisms (phagocytosis, edema formation etc.) in vivo and/or in vitro or by microglial depletion as a model to study the causal role of microglial inflammation.

3. The brain displays a complex and delicate interaction between all cell types of the brain. This also includes astrocytes, which are capable of a broad array of anti- and proinflammatory action after central nervous injury with heterogenous functional roles in disease and protection. The role of astrocytic involvement in the processes related to the describes protective preconditioning effect of hemorrhage should be addressed by the authors, for example by studying the impact of astrocyte co-culture experiments in vitro.

4. I think the authors should show the "negative" finding using the athymic nude and the NOD SCID mice, as these are important findings in the overall argument of T-cell involvement.

5. Figure 4: the authors first mention Figure 4C, then go back to Figure 4B and do not mention Figure 4A in the text at all. This should be corrected in the order of mention within the text.

6. Additionally, it remains unclear from the text or the figure legend whether the observed differences between the groups in Figure 4C are statistically significant.

7. Figure 4C: Furthermore, as the qPCR analyses in Figure 4C are done from whole brain tissue homogenates, the specificity of the observed changes in cytokine expression and whether this is solely due to microglia remains unclear. There will be a significant amount of astrocytes, other glia cells and neurons in the homogenates.

8. How was the relative comparison for the "delta-delta ct" method done exactly? What was the reference group for the "fold change" values. This is not clear from the methods or results section.

9. Statistics: The authors report that t-tests were used for the comparison of two groups. Were the data checked for normal distribution? As an alternative, Mann-Whitney test should be considered. More importantly, for Figure 1 and 3, repeated measures were done on individual mice over the time course of 1-7 days. Therefore, data should be analyzed using repeated measures ANOVA. For Figures 2, 4 and 5, there are more than two groups displayed on the graphs. It would be more appropriate to analyze the whole dataset per graph using one-way ANOVA.

10. Discussion: the translational applicability of the findings in the paper must be discussed in more depth: what could be a human correlate to the "pre mini stroke"? How could it be used therapeutically? Will it be possible to correlate the mouse data with data from humans (e.g. are patients with a history of small strokes less prone to severe clinical course after a major stroke)?

Reviewer #1 (Remarks to the Author):

This report is to document that a previous 'mini' ICH served as a protective role in a subsequent ICH, through enhancing T cell-dependent M2 microglia polarisation. This is an interesting topic.

There are some suggestions for the research team:

1 · Which sequence of MRI was used to evaluate the lesion? It should be described in the section of Method and materials.

Thank you for the important reminder. The missing information has been updated in Methods.

<Line 401-403> T2 weighted images were acquired as anatomical reference using a Rapid Acquisition with Refocused Echoes (RARE) sequence (FOV=32×32 mm², data matrix=256×256, RARE factor=8, TE/TR=36/4200ms).

2 · As the authors stated, the lesion volume includes hematoma, edema, and surrounding rim, so the volume reduction does not represent the hematoma clearance.

Indeed, 'hematoma clearance' was more suited for findings on histological brain sections, and it was a poor word choice for the subheading that also included MRI imaging results. We have since rephrased it as 'lesion resolution' in both Results and Figure Legends.

<Line 95-96> Lesion resolution occurred at a higher rate in mice with a previous stroke

<Line 672-673> (A) Coronal images, with maximal lesion area outlined in yellow, presented in chronological sequence. (B) Total lesion volume presented as percentage of initial lesion volume on day 1.

3 · The author attempted to prove the protective effect of previous 'mini'stroke is through enhancing M2 microglia polarisation. So the information of M2 microglia polarization and T cell activation after the first ICH is important.

Indeed, the action of microglia M1/M2 polarisation profile should be observed following the first mini-stroke. We have incorporated two new experimental groups of mice: the sham group and the mini-stroke only group both before and after the major hemorrhagic stroke. By doing so, we were able to trace microglia polarization profiles after the mini-stroke up until the day before the major hemorrhagic stroke took place. Both M1 and M2 percentages were measured via flow cytometry at days 1, 5, 9, and 13 after the first mini-stroke. We saw a decline to baseline levels at days 9 and 13, respectively, for M1 and M2. These findings were updated in Results, Methods, and Figure Legends 1 and 4B.

<Line 121-123> Both M1 and M2 levels returned to baseline in the sham and mini-stroke only groups before the major hemorrhagic stroke took place on day 0 (**Figure 4B**, gray lines)

<Line 345-346> Mice were randomly divided into four groups. The sham group underwent two sham operations on days -14 and 0. The 'past mini-stroke only' group underwent a single minor stroke on day -14.

<Line 685-687> Both M1 and M2 levels returned to baseline levels in the sham and mini-stroke only groups before the major hemorrhagic stroke took place on day 0 (Gray lines; n=4 on each day).

4 · It is better to provide two positive biomarkers to confirm microglia polarization.

In regards to newly-added Figure 4C, we have now verified microglia polarisation using CD163 and CD206 double staining on days 3, 5, and 7 for the control group and the previous stroke group. There is strong positive correlation in CD163 (also an M2 marker) and CD206, as demonstrated by a nearly identical percentage of double positive cells (Figure 4C) as when only CD206 was used in Figure 4B.

<Line 136-138> M2 polarisation was further verified with double staining of CD206 and CD163, which correlated with the main results (Figure 4C).

<Line 453-454> For M2 verification, cells were additionally stained with PE anti-mouse CD163 antibody, Biolegend #156703.

<Line 687-689> (C) Simultaneous expression of CD163 and CD206 revealed near-identical percentage of double positive cells on each day, further verifying M2 polarisation profile.

5 · Weather small-dose collagenase could induce mini-ICH ? Is the volume of hematoma stable in the first ICH? It is better to use the autologous blood injection model to control the hematoma volume.

We have indeed considered the blood injection model at the start of the experiment. While the mass effect of autologous blood and heme toxicity is similar to that of spontaneous ICH, the hematoma itself does not fully replicate a spontaneous hemorrhagic stroke in the human brain, as it lacks the downstream ischemic effects of ruptured small vessels, and the phenomenon of hematoma expansion as often observed clinically in the initial hours. Thereby, our hypothesis of ‘similar stimuli’ altering microglia polarization would be flawed if two different animal models were used for the mini and major strokes.

The collagenase model is a well-established experimental method to study spontaneous intracerebral hemorrhage. Units of collagenase have a good correlation with hematoma size, even at low doses. The key, we found, was to inject slowly (over a timeframe of 5-10 mins), and to lower the concentration for small units of collagenase, thus minimizing deviations of injection volumes. We have consistent histological evidence demonstrating similar mini-stroke volumes, which is presented as an added figure in the supplementary materials showing minimal inter-individual variations of the mini-stroke.

<Line 360-361> Replicable hematomas were induced using this dosage and method for the minor HS (Supplementary Figure 3).

Reviewer #2 (Remarks to the Author):

The study performed by Lee et al demonstrated that a past ‘mini-stroke resulted in better neurological function recovery, faster attenuation of hematoma and lesion volume, and better survival of hemorrhagic stroke mice, in a T cell-dependent manner. The study is interesting and could provide a therapeutic mechanism of hemorrhagic stroke.

1. The authors used major hemorrhagic stroke as a control group. However, I think they need to add a mini-stroke-only group and sham group in the study, to test if mini-stroke-only has any effects on brain injury and recovery.

Thank you for this important comment. We have incorporated two new experimental groups of mice: the sham group and the mini-stroke only group both before and after the major hemorrhagic stroke. By doing so, we were able to observe behavioral, body weight, and survival changes on days 1, 3, 5, and 7 after the major hemorrhagic stroke, as demonstrated in Figures 1A-E. Additionally, we were also able to trace microglia polarization profiles after the mini-stroke up until the day before the major hemorrhagic stroke took place (Figure 4B).

<Line 345-346> Mice were randomly divided into four groups. The sham group underwent two sham operations on days -14 and 0. The ‘past mini-stroke only’ group underwent a single minor stroke on day -14.

<Line 651-652> Sham group received two sham operations 14 days apart. Mice in the mini-stroke only group received a minor stroke on day -14.

2. In this study, M1 and M2 microglia are over-simplistic concepts of microglia. I would suggest not to use M1 and M2, but using pro-inflammatory microglia and anti-inflammatory microglia.

Indeed we agree that the dichotomy of M1/M2 is an over-generalization, especially as even within the M2 spectrum, there are M2a, M2b, and M2c subtypes, each expressing specific cellular markers. However, as M2-spectrum microglia not only induce anti-inflammatory effects, but also carry out actions such as erythrophagocytosis (M2a), hemoglobin clearance (M2b, 2c), and recruitment of Th2 and regulatory T cells (M2a), we believe that the term ‘M2’, although not perfect, still better represents the actions of the spectrum of cells we are observing. This has been updated in Discussion.

<Line 57-61> M2 microglia comprises of a wide spectrum of cells that promotes cell regeneration (M2a), phagocytosis of hemolytic products (M2b and M2c), and removal of cell debris (M2b and M2c). This shift in polarisation from M1 to M2 normally takes place on day 7.(9) Although the dichotomization of microglia into either M1/M2 phenotype is overgeneralized, for the sake of simplicity we would refer to them as such within this study.

<Line 216-223> Classically activated microglia demonstrate pro-inflammatory characteristics, and are labeled as ‘M1’-like phenotype, whereas alternatively activated ones were generalized as ‘M2’-like phenotype, comprising of M2a, 2b, and 2c subclasses, promoting cell regeneration, erythrophagocytosis, hemoglobin and cell debris clearance. Although the umbrella term ‘M2’ is not perfect, it adequately represents the spectrum of alternatively polarized cells we are observing. The dichotomy of activated microglia into M1 and M2 is admittedly an oversimplification, upon which, future research should be conducted to unpack the precise subtypes and their respective roles.

3. CD16/32, iNOS, CD206, Arginase are not specific markers of M1 and M2 microglia. I would suggest to use double immunostaining with IBA-1 to evaluate microglia polarization

We agree that a specific marker which remains positive throughout microglia lifespan, regardless of polarisation profile, should be used to identify cell population before talks of pro/anti-inflammatory action

could be debated. We regret that due to the logistics of the flow cytometry machine and antibody panel selection for specific immunofluorescence channels, we were not able to employ Iba-1 as an effective and consistent identifier for microglia. Rather, we chose CD45^{int}/CD11b^{high} double positive cells as microglia identifiers. This limitation of our study is now discussed under in Discussion.

On the other hand, IF staining has proven to be difficult to conduct in the early hours after a hemorrhagic stroke, due to massive hemoglobin interference.

To improve the validity of our findings, we have incorporated a new marker in our flow cytometry panel. By staining for CD163 and CD206 double positive cells, we confirmed that polarization towards anti-inflammatory microglia phenotype was well-grounded.

<Line 136-138> M2 polarisation was further verified with double staining of CD206 and CD163, which correlated with the main results (Figure 4C).

<Line 453-454> For M2 verification, cells were additionally stained with PE anti-mouse CD163 antibody, Biolegend #156703.

<Line 687-689> (C) Simultaneous expression of CD163 and CD206 revealed near-identical percentage of double positive cells on each day, further verifying M2 polarisation profile.

4. Besides microglia, macrophages will infiltrate into the brain after hemorrhagic stroke. What are the roles that macrophages played in mini-stroke induced protective effects?

Indeed, both bone marrow-derived macrophages and brain-derived microglia may contribute to the immune sequelae of hemorrhagic stroke. While our research focuses mainly on the latter, macrophages would almost certainly have infiltrated the brain parenchyma during our study. Several studies have thus umbrella-termed intracerebral CD11b⁺ cells as microglia/macrophages/monocytes. However, due to constraints of resources and time, we adopted a more focused approach with a more narrowly defined working hypothesis that only looks at microglia *in vitro*. The cells that were primary-cultured from

newborn pups were microglia-only, and yet they also demonstrated preferential M2 polarization and increased erythrophagocytosis after a simulated mini-stroke. Thus, while we could not refute nor confirm the role of macrophages in our described phenomenon, we were able to ascertain with reasonable certainty and specificity that microglia alone are at least partially responsible for eliciting the preferential polarization profile. <Line 207-214>

Reviewer #3 (Remarks to the Author):

In the manuscript entitled „A previous hemorrhagic stroke protects against a subsequent stroke via microglia alternative polarisation” the authors Lee et al. describe the interesting and novel observation that the modulation of microglial polarization via a previous hemorrhage can attenuate neuronal injury in a murine mouse model of hemorrhagic cerebral stroke.

Although the authors present some interesting results, I would like to raise a couple of points for improvement, especially concerning the complex cellular interaction within the brain and translational applicability. As a general opinion, the presented data present an excellent basis to report the interesting observation, but omit some important aspects that should be addressed:

1. The authors state a difference in survival depending on the pre-stroke intervention (Figure 1B). Was this difference statistically significant? This information is missing from the text or the figure legend.

Thank you for pointing this out. Figure 1B and the corresponding legends have been revised accordingly.

<Line 658> Survival curve of the previous stroke group and control group (n=25, Gehan-Breslow-Wilcoxon test, $p=0.0416$)

<Line 508-509> The Gehan-Breslow-Wilcoxon test was used in the Kaplan-Meier survival curve analysis.

2. The authors make the main argument that the modification of microglial polarization leads to improved neurological outcome due to faster hematoma clearance (Figure 2/3). But what is the mechanism of reduced hematoma size? Is it increased erythrophagocytosis and cell debris removal by microglia? Is it reduction of brain edema and blood-brain barrier disruption?

Is it really faster hematoma removal rather than less susceptibility to continued bleeding (“chronic” collagenase bleeding model, vascular stability)? Is it neuronal regeneration? Mechanical studies apart from inflammatory cytokine production are necessary to prove that microglial polarization (which the authors clearly show) is causal for the protective effects and not a consequence of unrelated changes in the microenvironment of the brain. This could be established by either investigating the above mentioned mechanisms (phagocytosis, edema formation etc.) in vivo and/or in vitro or by microglial depletion as a model to study the causal role of microglial inflammation.

(Revised in Methods, Results, and added Figure 6) In order to further evaluate the mechanism of the observed phenomenon *in vitro*, we conducted a RBC phagocytic study. Intra-microglial RBCs were quantified via flow cytometry using FITC-labeled RBC as the ‘major stroke’ trigger. The results, as presented within the revised manuscript, were quite promising. Data was compared between groups D, E, and F. A substantial increase in erythrophagocytosis response was seen in the group with a previous hemorrhagic stroke supplemented with T cell co-culture. As M2 microglia were characterized to present CD206, which is responsible for whole RBC phagocytosis, and CD163, which is responsible for hemoglobin clearance, it would be logical to conclude that increased RBC phagocytosis had an important role, albeit possibly a partial one, to play in hematoma clearance and lesion volume resolution.

<Line 182-186> When fluorescent labeled RBCs were added to microglia, the percentage of intracellular RBCs was highest in Group F (with T cells and a previous RBC coculture), second in Group E (with a previous RBC coculture) and lowest in Group D (RBC-naive microglia). (Figure 6) This indicates

superior RBC phagocytic abilities in microglia with a previous stimulus, which is enhanced by the presence of T cells.

<Line 493-499> Erythrophagocytosis study. Whole RBCs were labeled with PKH-26 fluorescent probe (Sigma-Aldrich) and co-cultured with primary microglia at a ratio of 10:1, as previously described.(33) After incubation at 37°C for 1 hour, the unengulfed RBCs were removed by washing thrice with PBS. Microglia were then harvested from the flask wall. The engulfed, intra-microglial RBCs were detected by flow cytometry via the FITC channel. The level of phagocytosis was quantified by the positive FITC percentage of each measurement.

<Line 718-724> Fig 6. Erythrophagocytosis study on primary cultured microglia in vitro. PKH-26 labeled whole RBCs were cocultured with microglia for 1 hour before being removed by PBS washing. Engulfed fluorescent RBCs remained within the flask-adherent microglia and were analyzed via flow cytometry with the FITC channel. a) Phagocytic activity was highest in Group F (previous RBC coculture with T cells), second highest in Group E (previous RBC coculture) and lowest in Group D (no previous RBC coculture). Error bars indicate SEM. * $p < 0.05$, ** $p < 0.01$. b) Graphical representation of RBC-FITC+ percentage.

3. The brain displays a complex and delicate interaction between all cell types of the brain. This also includes astrocytes, which are capable of a broad array of anti- and proinflammatory action after central nervous injury with heterogenous functional roles in disease and protection. The role of astrocytic involvement in the processes related to the describes protective preconditioning effect of hemorrhage should be addressed by the authors, for example by studying the impact of astrocyte co-culture experiments in vitro.

Indeed, the multitude of cell types and complex environment of the brain renders it difficult to attribute an observed phenotype to a single group of cells. However, due to constraints of resources and time, we chose to adopt a more focused approach with a more narrowly defined working hypothesis that only looks

at microglia *in vitro*. While a microglia-only *in vitro* approach was able to demonstrate preferential M2 polarization and an enhanced erythrophagocytic response (which enabled us to ascertain that microglia alone were at least partially responsible for eliciting the observed phenotype), astrocytes were most certainly present during our study. The specific role of which astrocytes play, possibly in complex interactions with microglia due to their broad array of anti- and pro-inflammatory actions would be a promising direction for future studies, which has been added to our Discussion.

<Line 320-322> Astrocytes and oligodendrocytes also secrete chemokines such as CCL2, CXCL1, and CXCL10, which might have roles in microglial differentiation and could be explored in future studies.(9)

4. I think the authors should show the “negative” finding using the athymic nude and the NOD SCID mice, as these are important findings in the overall argument of T-cell involvement.

The phenotypic observations of athymic nude mice have been updated in our supplementary materials.

<Line 113> Interestingly, there was no difference in phenotypical results between the previous stroke groups and the control groups (Supplementary Figure 1).

5. Figure 4: the authors first mention Figure 4C, then go back to Figure 4B and do not mention Figure 4A in the text at all. This should be corrected in the order of mention within the text.

This mistake has been revised in Results. In addition, Figure 4C has now been changed to CD206/CD163 double staining, whereas Figure 4D is now qPCR analysis results.

6. Additionally, it remains unclear from the text or the figure legend whether the observed differences between the groups in Figure 4C are statistically significant.

The statistical description has been added to the figure legend of figure 4D (originally figure 4C.)

<Line 695-696> The results of qPCR analysis were not statistically significant. Error bars indicate SD.

7. Figure 4C: Furthermore, as the qPCR analyses in Figure 4C are done from whole brain tissue homogenates, the specificity of the observed changes in cytokine expression and whether this is solely due to microglia remains unclear. There will be a significant amount of astrocytes, other glia cells and neurons in the homogenates.

Indeed, whole brain homogenate has its shortcomings when being used as a source of qPCR sample collection. An array of potential interference could be present, including necrotic or apoptotic neurons, astrocytes, and oligodendrocytes. Perhaps this is the reason behind a lack of significance in regards to qPCR results. This has been added to our Discussion.

<Line 312-315> qPCR analysis was performed on whole brain homogenates, which includes not only microglia but also an array of astrocytes, neurons, necrotic tissue and cell debris. Unlike flow cytometry, qPCR results did not show statistical significance. This could possibly be attributed to the wide range of interference from the aforementioned components.

8. How was the relative comparison for the “delta-delta ct” method done exactly? What was the reference group for the “fold change” values. This is not clear from the methods or results section.

This missing information has been updated in Methods.

<Line 419-421> The threshold cycle (CT) was normalized to GAPDH (ΔCT) of the same mouse and then to that of the sham mouse ($\Delta\Delta\text{CT}$). Mean fold changes are depicted as $2(-\Delta\Delta\text{CT})$. Data acquisition was conducted in technical triplets.

9. Statistics: The authors report that t-tests were used for the comparison of two groups. Were the data checked for normal distribution? As an alternative, Mann-Whitney test should be considered. More importantly, for Figure 1 and 3, repeated measures were done on individual mice over the time course of 1-7 days. Therefore, data should be analyzed using repeated measures ANOVA. For Figures 2, 4 and 5, there are more than two groups displayed on the graphs. It would be more appropriate to analyze the whole dataset per graph using one-way ANOVA.

The new data analysis tests has been updated in the Methods section, along with the figure legends of corresponding graphs.

<Line 501-509> Phenotypical studies that included multiple measurements of the same mouse over time (body weight, mNSS, Rotarod, MRI total lesion volume) were analyzed by repeated-measurements two-way ANOVA with matched values of the same mouse across time. Additionally, Sidak's multiple comparisons test were conducted on the mean value obtained on each date between experimental groups. Mechanistic studies that require sacrificing mice at time of measurement (flow cytometry, qPCR) were analyzed by regular two-way ANOVA (no repeated measures) and Sidak's multiple comparisons test. Cell line studies involving more than two groups of importance to compare were analyzed by one-way ANOVA with Tukey's multiple comparisons test. The Gehan-Breslow-Wilcoxon test was used in the Kaplan-Meier survival curve analysis.

<Line 659-661> (c) Measurements of body weight (n=10, RM two-way ANOVA, p=0.0183). (d) Modified neurological severity score assessment (n=10, RM two-way ANOVA, p= 0.0003). (e) Rotarod testing. Results presented as latency to fall (seconds) (n=10, RM two-way ANOVA, <0.0001).

10. Discussion: the translational applicability of the findings in the paper must be discussed in more depth: what could be a human correlate to the “pre mini stroke”? How could it be used therapeutically? Will it be possible to correlate the mouse data with data from humans (e.g. are patients with a history of small strokes less prone to severe clinical course after a major stroke)?

Indeed, although our study is merely pre-clinical in nature, the potential application to human equivalents can be multifold. This has been further elaborated in Discussion. We have previously strived to collect clinical data on patients with two separate hemorrhagic strokes. However, documentation was scarce on minor bleeds, either due to misclassification of subclinical minor hemorrhagic strokes as ischemic strokes or transient ischemic attacks, or due to the low incidence of a minor hemorrhagic stroke happening in isolation yet still presenting as an emergent hospital admission warranting extensive brain imaging. As of

now, the best approach for future studies is perhaps retrospective, incidental discoveries of scarring on brain imaging to indicate that a past mini-stroke has happened in the human brain. Yet even then, it may be hard to differentiate between past ischemic or hemorrhagic types.

<Line 291-302> The human correlate of a mini-stroke can take many forms, such as introduction of small amounts of whole RBCs into the CSF via lumbar puncture which then induce dormant microglia memory within the central nervous system as a form of active immunity. Another approach is through passive immunity in the form of *ex vivo* cell therapy, such as introducing HLA-matched, trained microglia (or T cells) into the CSF within the early hours after a hemorrhagic stroke has occurred. Lastly, if the antigen of HS presented on microglia could be identified and sequenced via proteomics, perhaps by combining it with microglia MHC-II, the end product would be able to induce immune memory formation to combat an upcoming, major hemorrhagic stroke or possibly other forms of intracranial hemorrhage such as traumatic hematomas. The overarching approach for future studies is to exploit the brain's innate immune response to a blood-specific trigger, introduced in a clinically benign way, so as to enhance the body's capabilities to deal with an actual clinical event.

REVIEWERS' COMMENTS:

Reviewer #1 (Remarks to the Author):

The authors have responded to all comments and provided necessary supplemental results to confirm their conclusion. Therefore, I agree to accept this manuscript.

Reviewer #2 (Remarks to the Author):

The authors have addressed all my concerns. I suggest the manuscript to be accepted.

Reviewer #3 (Remarks to the Author):

The authors present a revised version of the manuscript entitled "A previous hemorrhagic stroke protects against a subsequent stroke via microglia alternative polarisation".

The revision was thoroughly done. Questions and concerns have been addressed either by additional experiments or adequate additions to the discussion.

I would like to raise one single aspect:

1. Please mention the fact that the qPCR data (Figure 4D) does not show significant differences between the group within the results description and not only in the figure legend.